# Anti-Idiotypic Nanobodies Mimicking an Epitope of the Needle Protein of the Chlamydial Type III Secretion System for Targeted Immune Stimulation

**DOI:** 10.3390/ijms25042047

**Published:** 2024-02-07

**Authors:** Ekaterina A. Koroleva, Oksana S. Goryainova, Tatiana I. Ivanova, Marina V. Rutovskaya, Naylia A. Zigangirova, Sergei V. Tillib

**Affiliations:** 1Institute of Gene Biology of the Russian Academy of Sciences, Vavilova Str. 34/5, 119334 Moscow, Russia; korolevakate@yandex.ru (E.A.K.);; 2National Research Center for Epidemiology and Microbiology Named after the Honorary Academician N. F. Gamaleya, 123098 Moscow, Russia; 3Engelhardt Institute of Molecular Biology of Russian Academy of Sciences, Vavilova Str. 32, 119991 Moscow, Russia

**Keywords:** anti-idiotypic antibodies, single-domain antibody, nanobody, *Chlamydia*, urogenital infection, immunomodulation, biomimetic

## Abstract

The development of new approaches and drugs for effective control of the chronic and complicated forms of urogenital chlamydia caused by *Chlamydia trachomatis*, which is suspected to be one of the main causes of infertility in both women and men, is an urgent task. We used the technology of single-domain antibody (nanobody) generation both for the production of targeting anti-chlamydia molecules and for the subsequent acquisition of anti-idiotypic nanobodies (ai-Nbs) mimicking the structure of a given epitope of the pathogen (the epitope of the Chlamydial Type III Secretion System Needle Protein). In a mouse model, we have shown that the obtained ai-Nbs are able to induce a narrowly specific humoral immune response in the host, leading to the generation of intrinsic anti-*Chlamydia* antibodies, potentially therapeutic, specifically recognizing a given antigenic epitope of *Chlamydia*. The immune sera derived from mice immunized with ai-Nbs are able to suppress chlamydial infection in vitro. We hypothesize that the proposed method of the creation and use of ai-Nbs, which mimic and present to the host immune system exactly the desired region of the antigen, create a fundamentally new universal approach to generating molecular structures as a part of specific vaccine for the targeted induction of immune response, especially useful in cases where it is difficult to prepare an antigen preserving the desired epitope in its native conformation.

## 1. Introduction

*Chlamydia trachomatis* is the most common sexually transmitted bacterial pathogen. It causes serious health problems in humans and can cause serious complications, such as pelvic inflammatory disease, ectopic pregnancy and infertility in women [1]. The medical and socio-economic significance of the search for new-generation drugs using target-specific technologies is due to the lack of effective agents that can help treat chronic bacterial infections and the rapid development of pathogenic resistance to the antibacterial drugs used to treat acute infectious diseases [2,3,4]. In the case of antibacterial drugs, this technology involves selecting as targets the proteins responsible for the microorganism’s display of pathogenic properties, then searching for specific inhibitors using computer software, organic synthesis methods and experimental studies and validating the predicted biological activity on model systems of the infectious process. The secretion of pathogenic factors (proteins responsible for the manifestation of pathogenic properties in bacteria) into macroorganisms’ cells is the key mechanism underlying the development of the infectious process. The type 3 secretion system (T3SS) is the predominant virulence factor of chlamydiae. It is essential for cell invasion and is active at all life stages [5,6]. Some T3SS proteins are on the surface and can be targeted by neutralizing antibodies. A T-cell response to T3SS antigens has recently been shown to be associated with protection against *C. trachomatis* infection in highly exposed women [7], and T3SS components have recently attracted attention as vaccine candidates against other pathogenic bacteria [8,9,10,11]. The *C. trachomatis* T3SS filament protein CdsF and its orthologs in other bacteria, such as TC_0037 protein of *Chlamydia muridarum*, form the needles of injectisomes and are believed to facilitate the insertion of translocators into the host cell membrane [5,6,12]. CdsF (TC_0037) is highly conserved, showing 95% sequence identity in the genus *Chlamydia*. It is abundant on bacterial surfaces, raising the possibility that a CdsF-based vaccine may induce a wide range of protection against all medically significant strains. It was shown that the T3SS needle protein TC_0037 induced specific humoral and T-cell responses, decreased the *Chlamydia* loads in the genital tract and abrogated the pathology of the upper genital organs. It could be a good candidate for inclusion in a *Chlamydia* vaccine [13]. In the study just mentioned, the authors immunized mice with a replication-defective adenoviral vector expressing the recombinant target antigen TC_0037. Thus, TC_0037, based on the published data, is a very promising target for the immunotherapeutic suppression of chlamydial infection. The presented work was made possible due to the fact that at the previous stage of research, we were able to obtain highly specific single-domain antibodies (nanobodies) that recognize precisely this promising therapeutic target, TC_0037.

The purpose of our study was to demonstrate the fundamental possibility of using another approach of transmitting information about the target antigen to the host body for targeted immune stimulation. The approach is based on the use of stable molecular structures that mimic the specific pathogenic epitope that is the preferred target of a therapeutic antibody. One of the ways to create such biomimetics is based on the production of anti-idiotypic antibodies (aiAbs). An “idiotype” is an “individual antigenic determinant” for an isolated antibody or for a specific type of T- or B-lymphocyte receptor [14]. According to the modern terminology, originally proposed in 1960 by N. Yerne [15], epitopes are determinants of antigens that are recognized by a specific antibody-binding site, the paratope. An idiotype is a structure formed by the association of the variable regions of the heavy and light chains of an antibody. Each idiotype is capable of stimulating the production of complementary “anti-idiotypic” antibodies, the antigen-recognizing site of which may be conformationally similar to the original antigenic determinant (a specific epitope of the antigen). Thus, the active centers of the antibodies formed in response to antigen introduction, in turn, act as antigens. According to the anti-idiotypic principle, the immune system is potentially capable of reproducing or at least modifying the action of any biologically active endogenous or exogenous immunogenic agent, as well as probably regulating a wide range of cellular functions in the body. aiAbs are used when it is difficult to prepare and apply an antigen or when the antigen is toxic, unstable or causes an insufficiently focused immune response. Finally, the native structure of the antigen may be poorly understood, and antibodies (idiotypes) that specifically recognize it in vivo have already been obtained. Due to the potential ability of carefully selected aiAbs to imitate a certain epitope (display the “internal image”) of a given antigen, thus compete with the antigen in binding to the same structures in living systems and competitively change the biological activity of the antigen, aiAbs have become a promising tool in the search for new ways to treat autoimmune diseases, cancer and many other human diseases [16,17]. It was demonstrated that aiAbs may very likely have therapeutic potential for use as substitutes (mimetics) of antigens (both of protein, non-protein or mixed nature) in vaccines for the treatment of bacterial, viral and cancer diseases [18,19,20].

In our opinion, single-domain antibodies (nanobodies) are a very promising tool for obtaining such molecules, anti-idiotypic nanobodies (ai-Nbs), structurally mimicking a given epitope [21,22,23,24,25]. It was shown that ai-Nbs can be effective mimetics of natural enzymes, an excellent tool for studying and utilizing molecular mimicry techniques. ai-Nbs are easier to generate and better suited to mimicking small ligands and to the subsequent design of peptide mimetics for drug development [26,27,28,29]. In camelids, an unusual combination of two types of antibodies/receptors of B-cells with markedly different properties was found. The repertoires of possible paratopes of the antigen-binding structures of heavy-chain-only antibodies (HCAbs) and classical antibodies, apparently, may differ markedly. Since these two types of antibodies co-exist in the same organism, it can be assumed that they do not compete but mutually complement each other. Indeed, it has been repeatedly observed that both types can arise in parallel (mutually exclusive or in different ratios) against different epitopes/antigens of multicomponent antigenic material used in the immunization of the same animals. Some authors have expressed doubts about the possibility of adequate antigenic mimicry of the usually convex epitopes of antigens with the help of classical (albeit anti-idiotypic) antibodies, the active centers of which usually have a concave structure [25]. However, such a protruding structure is quite realistic in the case of non-canonical HCAbs (and their derived nanobodies), in which the more elongated third hypervariable region (CDR3) forms exactly protruding structures [24,30,31,32]. Thus, the camelid arsenal has the potential to create “more complementary” interactions within the idiotype–anti-idiotype network. In our opinion, it is the structural features of single variable domains (VHH) that make nanobodies a potentially unique tool for creating an anti-idiotype that mimics a specific epitope of the target antigen in terms of specific binding.

In this study, we applied nanobody generation technology to obtain ai-Nbs mimicking the structure of a given model epitope of the pathogen (the epitope of the Chlamydial Type III Secretion System Needle Protein). In a mouse model, we show that the obtained ai-Nbs are able to induce a narrowly specific humoral immune response of the host leading to the generation of intrinsic anti-*Chlamydia* antibodies, potentially therapeutic, specifically recognizing a given antigenic epitope of *Chlamydia*. The immune sera derived from mice immunized with ai-Nbs are able to partly suppress chlamydial infection in a neutralization assay in vitro. We hypothesize that the use of ai-Nbs, which mimic and present to the host immune system exactly the desired region of the antigen, creates a fundamentally new universal approach to generating molecular structures as a part of a specific vaccine for the targeted induction of immune response, especially useful in cases where it is difficult to prepare an antigen preserving the desired epitope in its native conformation.

## 2. Results

### 2.1. The Selection and Modification of an Initial Nanobody aChlNP Recognizing the Chlamydial Type III Secretion System Needle Protein

Earlier, we described the preparation of a number of promising nanobodies that effectively bind to extracellular and intracellular forms of *Chlamydia trachomatis*, as well as have activity that inhibits the development of chlamydial infection under in vitro conditions [33]. We were unable to identify with which particular *Chlamydia* antigen the obtained nanobodies bind. In Western blot analysis, we did not see the target proteins and assumed that the native conformational epitopes recognized by the obtained nanobodies may have been disrupted during protein isolation and denaturation or that the target antigens may have been poorly soluble, aggregated or been lost during protein extraction. Already after the publication of the aforementioned article, we performed additional refined selection procedures (using the phage display method) using the recombinant protein TC_0037, produced in *E. coli* bacteria, corresponding to the conserved type 3 secretion system (TTSS) protein TC_0037 of *C. muridarum*, as the antigen for selection. In this case, we used the immune library of the nanobody clones described in our article, which was obtained by cloning the entire repertoire of variable domains (nanobodies) of specific camel antibodies, consisting of a homodimer of shortened heavy chains in the absence of light chains, from the peripheral blood lymphocytes of a camel immunized with a preparation of native purified *C. trachomatis* elementary bodies. Both whole bacterial cells of *C. trachomatis* (strain Bu-434), inactivated by UV irradiation, and a protein complex without LPS, prepared from the outer membrane of the chlamydia cell wall, were used as antigens for the immunization. As a result of this work, we were able to obtain one particularly promising nanobody (aChlNP, Figure 1) that recognized the TC_0037 protein (the “*Chlamydia* injectisome needle” protein) while exhibiting very specific binding to chlamydiae, including intracellular chlamydiae, forming inclusions (similar staining for the two originally derived nanobodies, as well as all methodological procedures, was described earlier [33]). The initially selected coding sequence of the aChlNP nanobody was further formatted according to the traditional protocol of our laboratory, as described previously [34]. In this case, a linker region, HA-tag and His-tag (to make aChlNP-HH) were added to the C-terminus of the nanobody. To obtain the nanobody in a trimerizable format, a trimerizing isoleucine zipper (ILZ) domain sequence was inserted after the linker region as described in [34]. The formatted aChlNP sequence is shown in Figure 1a. The sequence data have been submitted to the GenBank database under accession number OR885932. The formatted aChlNP with the trimerizing ILZ domain, (aChlNP-ILZ-HH)x3, was expressed in E. coli and purified via His-tag affinity chromatography. Before analysis using SDS-PAAG (Figure 1b), the trimerized nanobody ChlNP was additionally purified from small impurities and, presumably, from monomeric nanobodies using ultrafiltration on a filter with a MWCO of 50 kDa (Vivaspin Turbo 4, Sartorius Stedim Lab Ltd., Stonehouse, UK). This nanobody remained almost entirely at the top and did not pass through this filter. Reactivity tests using ELISA revealed that the purified nanobody specifically recognizes the recombinant TC_0037 protein (Figure 1c). The trimerizing variant of aChlNP is noticeably more effective in ELISA, as expected. The nanobody aChlNP at a concentration of 5 μg/mL specifically bound antigens of closely related *Chlamydia* species, demonstrated using a micro-immunofluorescence assay on eukaryotic McCoy cells infected with *C. trachomatis* or *C. muridarum* (Figure 1d). The inhibitory effect of the aChlNP nanobody at concentrations of 10 μg/mL preincubated with *C. trachomatis* elementary bodies on their intracellular development in vitro was demonstrated [33]. We found this aChlNP nanobody in the trimerizing format to be the most promising for use as an idiotype for the creation of anti-idiotypic nanobodies (ai-Nbs) mimicking an epitope of the Chlamydial Type III Secretion System Needle Protein. 

### 2.2. Camel Immunization with aChlNP and ai-ChlNP Nbs Generation and Selection

The formatted aChlNP nanobody was used for the immunization of a new Bactrian camel to generate ai-Nbs that specifically recognize the antigen-binding site (idiotype) of the aChlNP. The ai-Nbs finally selected by phage display must be tested for the possibility of their use through immunization to induce formation in the body of experimental animal (mouse) of its own host’s antibodies that specifically recognize the primary antigenic epitope of Chlamydia injectisome needle protein, ChlNP. 

The finally selected anti-idiotypic nanobodies should be tested for the possibility of their use by means of immunization to induce the formation in the body of experimental animal (mouse) of antibodies that specifically recognize the primary antigenic epitope of Chlamydia injectisome.

The immunization procedures, the cloning of the entire repertoire of variable domains of specific antibodies from the immunized camel lymphocytes and the subsequent selection of nanobodies were performed as described previously [34]. The camel immunization was performed according to a standard protocol, using a mixture of approximately 1 mg of aChlNP nanobody with an equal volume of LQ adjuvant (GERBU, Germany) for each step (Figure 2a). We observed using ELISA a marked increase in the antibody titer with a given specificity as a result of the immunization (Figure 2b). On the basis of the mRNA isolated from the peripheral blood lymphocytes of the immunized camel, using pairs of specially selected primers for PCR, we synthesized and cloned in a phagemid vector (pHEN4) the whole repertoire of cDNAs encoding the variable antigen-recognizing regions (“single-domain antibodies”) of the camel-specific antibodies, consisting only of a dimer of shortened heavy chains. From the obtained cDNA library, we performed functional selection using modified phage display of the single-domain antibody clones that specifically bind to the antigen-recognizing site (idiotype) of the aChlNP nanobody but do not bind to the constant sites of these nanobodies (we used other nanobodies available in our lab of the same general framework structure, but with a different recognition specificity, for negative and competitive selection). As a result of the selection procedures, we were able to obtain four different variants of new single-domain antibodies specifically recognizing the aChlNP nanobody idiotype (anti-idiotypic nanobodies, ai-Nbs, aiChlNP).

The sequences of four selected ai-Nbs (aiChlNP-74, aiChlNP-75, aiChlNP-79 and aiChlNP-81) are shown in Figure 3. The sequence data have been submitted to the GenBank database under accession numbers OR901958, OR901959, OR901960 and OR901961, correspondingly. The formatted (aiChlNP-HH) ai-Nbs were expressed in *E. coli* and purified (Figure 3c).

### 2.3. AiChlNP Induce the Formation of IgG Antibodies Specific to the TC_0037 Protein of C. muridarum 

These four selected and formatted ai-Nbs (aiChlNP-74, aiChlNP-75, aiChlNP-79 and aiChlNP-81) were used in the following experiments of mice immunization aimed to verify whether these ai-Nbs would be able to induce the formation of anti-chlamydial IgG antibodies in the host organism.

After ai-Nbs immunization, sera from mice were obtained and analyzed for the presence of antibodies specific to the recombinant TC_0037 protein (rTC_0037). As shown in Figure 4a, four-stage immunization with ai-Nbs resulted in the formation of antibodies specific to the chlamydial protein TC_0037. The highest IgG antibody titers were shown for the samples aiChlNP-74, aiChlNP-75 and aiChlNP-79 relative to the intact serum, where no antibodies were formed. In the case of immunization with the primary aChlNP or trimerized nanobodies of a different specificity (nanobodies against the hemagglutinin of the influenza A virus [34] as a control, control Nbs), we, similarly to in the intact control serum, did not observe any increase in the titer of mouse antibodies binding to the chlamydial antigen TC_0037.

Next, we analyzed the combined effects of ai-Nbs in three-stage immunizations of mice. Since we do not yet know the features of either the dynamic nature of the recognizable epitope or the characteristics (for example, the degree of stabilization of the paratope loops) of specific selected ai-Nbs, one could assume the possibility of a synergistic effect of using a combination of the obtained ai-Nbs to induce in the host body the generation of corresponding antibodies with slight variations in their paptopes, which theoretically could collectively improve their binding to the target pathogenic epitope. This work did not reveal a significant synergistic effect. We reduced the scheme of ai-Nbs administration, as we had used several ai-Nbs at once, and compared that with only aiChlNP-74, which showed the highest titers of IgG antibodies specific to the rTC_0037 protein according to the ELISA data (Figure 4a). Sera from intact mice were used as the controls. Immunization with ai-Nbs in combination (aiChlNP-74 and -81; and aiChlNP-74, -75, -79 and -81) induced the production of TC_0037-specific antibodies at a titer of 1:12,800, while the difference in titers compared to the sera obtained via immunization with aiChlNP-74 alone was insignificant, which once again confirms the high immunogenicity of aiChlN-74 (Figure 4b).

### 2.4. Immune Sera Derived from Mice Immunized with ai-Nbs Are Able to Partly Suppress Chlamydial Infection in a Neutralization Assay In Vitro

Four anti-ai-Nb immune sera samples (anti-aiChlNP-74, -75, -79 and -81) and intact control or anti-control Nb immune sera samples were tested for their ability to neutralize chlamydial infection in vitro. The antibodies resulting from immunization with aiChlNP-74, -75 and -81 showed the highest neutralizing activity at the lowest dilution (1:32) with an average of 70.8% for aiChlNP-74, 60.8% for aiChlNP-75 and 69.75% for aiChlNP-81, compared to the pre-immune sera (Figure 5). At a final titer of 1:528, the percentage for the nanobodies (aiChlNP-74, -75, -79 and -81) averaged 10.47% neutralization of the chlamydial infection. The two most promising variants of the anti-idiotypic nanobodies, aiChlNP-74 and aiChlNP-81, were identified in this test.

### 2.5. Immunization of Mice with ai-Nbs (aiChlNP-74, -75, -79 and -81) Induces the Formation of IgG1 and IgG2a Antibodies Specific to the C. muridarum TC_0037 Protein

Interferon-γ (IFN-γ), as a Th1 (type 1 T-helper cell) cytokine, and interleukin-4 (IL-4), as a Th2 cytokine, are known to induce isotype switching to IgG2a and IgG1, respectively. Moreover, higher IgG1 titers are usually associated with disease progression, and in contrast, higher IgG2a titers are related to protection from disease. In this study, we evaluated using ELISA the extent of the induction of specific antibodies of IgG1 and IgG2a isotypes in response to immunization with the ai-Nbs. Our data show that in animal sera, in response to immunization with aiChlNP-74, -75 and -81, specific antibodies of the IgG2a and IgG1 isotypes are detected. The IgG2a/IgG1 ratio of specific anti-TC_0037 antibodies, determined in sera diluted 1:100 as the average of six OD450 (IgG2a)/OD450 (IgG1) measurements (+SE), was greater than 1 in case of all three ai-Nbs, which may indicate a differentiated immune response predominantly to type 1 T-helper cells (Figure 6). This result indicates the efficacy of the selected ai-Nbs and allows one to consider aChlNP-74, -75 and -81 as potential candidate molecules for future study of their therapeutic anti-infective activity.

### 2.6. aiChlNP Immunization Induces the Secretion by T-cells of IFN-γ, TNF-α and IL-2 in Mice

Since IFN-γ has been found to play an important role in mediating control over chlamydial infection, we evaluated the IFN-γ production by the T-cells in response to UV-killed *C. muridarum* and the rTC_0037 protein in ai-Nb-immunized mice. The T-cell responses specific to TC_0037 and *C. muridarum* were evaluated using ELISPOT two weeks after the last immunization. As shown in Figure 7, stimulation of the UV-killed *C. muridarum* and rTC_0037 induced strong IFN-γ production by the T-cells derived from the mice immunized with the aiChlNP-74 and aiChlNP-81 nanobodies. The IFN-γ induction in response to immunization with aiChlNP-75 was lower relative to with the aiChlNP-74 and -81 nanobody series but was significantly higher than the level of IFN-γ induction in the mice immunized with an adjuvant alone and the negative control (intact mice). It is worth noting that the production of a specific IFN-γ T-cell response during immunization with aiChlNP-74 and aiChlNP-81 in response to the stimulation of the splenocytes with rTC_0037 was higher in comparison with the group of mice infected with *C. muridarum*, indicating the high specificity of the obtained antibodies. Interestingly, when the splenocytes were stimulated with the UV-inactivated *C. muridarum*, the IFN-γ production in response to immunization with aiChlNP-74 and aiChlNP -81 correlated with the response in the mice infected with *C. muridarum* (natural infection), suggesting that these nanobodies may have some therapeutic activity (Figure 7a). 

Another significant indicator of immune response, especially for controlling intracellular infection, is TNF-α. We assessed the ability of T-cells stimulated with chlamydial antigens to induce TNF-α production (Figure 7b). We did not observe an inflammatory response when the cells were stimulated with the protein rTC_0037, but the T-cells actively produced TNF-α in response to stimulation with the inactivated *C. muridarum* (Figure 7b). As with a natural infection, we obtained a pro-inflammatory response in mice to *C. muridarum* infection. TNF-α in combination with IFN-γ can synergize to mediate killing pathogens [35]. 

IL-2, unlike IFN-γ and TNF-α, has no effector function, but strongly enhances the expansion of the effector T-cells. Therefore, when assessing the inflammatory response, it is important to evaluate the significance of this cytokine. It has been shown that during chlamydial infection, the level of IL-2 production is much higher in response to stimulation with inactivated *C. muridarum* than in response to the rTC_0037 protein. But at the same time, we observe that immunization with the ai-Nbs, especially aiChlNP-75 and -81, induces almost the same level of IL-2 as natural infection (Figure 7c). This suggests that the inflammatory response observed after immunization with the studied ai-Nbs, with the increased proliferation of the effector cells due to the production of IL-2, will provide more effective protection against *C. muridarum* infection.

IL-6 is important in the immune defense against chlamydial infection. IL-6 is required to control chlamydial infection by limiting pathogen replication and colonization at both high and low doses of Chlamydia muridarum. In addition to its role in host defense, IL-6 also mediates inflammatory pathology [36]. Like TNF-α, IL-6 is actively produced in response to *C. muridarum* infection, but we have shown that in response to stimulation with *Chlamydia* antigens, we did not observe a significant level of IL-6 production, which excludes the development of pathological changes as a result of immunization with ai-Nbs.

As for the induction of the anti-inflammatory cytokines IL-4 and IL-10, we did not observe a high level of their production by the splenocytes. They were approximately at the same level as in the intact mice, <100 SFU, which correlates with the pro-inflammatory cytokine data. 

## 3. Discussion

The main goal of this study was the application of nanobody generation technology to the methodological development and demonstration, as a proof of principle, of an efficient way to create structurally stable polypeptide biomimetics that conformationally reproduce a given epitope of a target pathogen, able to present to the host immune system exactly the desired region of the antigen for the targeted induction of immune response. In our opinion, we managed to achieve this goal. The method of producing the nanobodies described in this article was used for the first time, as far as we know. The main proposed stages for obtaining anti-idiotypic nanobodies are as follows.

(1) With the described method, it is fundamentally important to identify a promising therapeutic target at the beginning of the work and to be able to obtain primary single-domain antibodies that highly specifically recognize it and have therapeutic potential. Most likely, this will be an already known target for which monoclonal antibodies (mAbs) have been obtained and the immunotherapeutic effect of using these antibodies has been demonstrated. 

(2) The generation of primary nanobodies against a selected target. Here, we have a classic case of obtaining new nanobodies using a well-established technological platform. From the panel of obtained nanobodies, it is necessary to functionally select those variants that effectively bind to the selected target in vivo and have the desired effect (for example, they can suppress the infectivity of *Chlamydia* in vitro).

(3) The functionally selected primary nanobody must be formatted in such a way as to most effectively use it as an antigen for the subsequent immunization of a member of the *Camelidae* family (using special formatting, carrier proteins and effective adjuvants). From our experience, a parallel trimerized version of the nanobody reproducibly gave a good result in inducing the formation of ai-Nb precursors when immunizing a camel. The use of special formatting to obtain trimerizing nanobodies is a development of our laboratory [34].

Among some other published examples of the use of anti-idiotypic nanobodies, which are still rare but show great potential, ai-Nbs are usually produced against the idiotypes of classical antibodies [26,27,28,29,37,38,39]. We consider this an alternative possible route; however, in our experience, the efficiency of the generation and selection of ai-Nbs that mimic the epitope of the original target increases significantly if nanobodies are used as the primary idiotypes instead of mAbs.

(4) The generation of ai-Nbs using a well-established technological nanobody platform. The main desirable criteria for the selection of ai-Nb clones at this stage are: (a) the high specificity and affinity of the binding of selected ai-Nb variants to a unique idiotype (paratope) and not to conservative regions of the primary nanobody and (b) the ai-Nbs should effectively compete with the original target antigen (in this case, TC_0037) in binding to the primary nanobody.

(5) Selected ai-Nbs must be formatted in such a way as to most effectively use them as antigens for the subsequent immunization of a model organism (mouse). For the initial demonstration experiments (in this article), we also used a trimerized version of the ai-Nbs, but this is only one of the possible formats for preparing antigenic material. 

(6) Immunization of a model animal (mouse) with antigenic material based on a formatted ai-Nb (or its multi-module derivatives) and functional analysis of the induced host antibodies. The goal is to select an ai-Nb that is able to induce a narrowly specific humoral immune response in the host leading to the generation of intrinsic antibodies against the initially selected therapeutic target.

With the described method, it seems fundamentally important to identify a promising therapeutic target at the beginning of the work and to be able to obtain primary single-domain antibodies that highly specifically recognize it and have therapeutic potential. At the beginning of the presented work, we were able to obtain the desired highly specific single-domain antibody (nanobody) that recognized precisely such a promising therapeutic target, Chlamydial Type III Secretion System Needle Protein TC_0037.

It can be stated that the following sequence of events was triggered: antigen X → nanobody aX recognizing X → anti-idiotypic nanobody (anti-aX) recognizing nanobody aX → induction of the formation of mouse antibodies recognizing antigen X, where the arrow denotes the immunization step. First, a camel (or another member of the Camelidae family) is immunized with biological material enriched with the target antigen in a conformation as close as possible to the native one. Using the RNA of the peripheral blood mononuclear cells of the immunized camel, molecular cloning of the entire repertoire of variable domains (VHH, nanobodies) of the special antibodies (HCAbs) is carried out, and the most promising clone variants are selected using the phage display method. After functional testing, the best variants of the primary nanobodies (aChlNP) are selected for the second cycle of immunization, also using a representative of the Camelidae family. From our previous experience, we hypothesized that the primary nanobody in a trimerizable format would be a more effective inducer of the desired immune response. To this end, during formatting, additional amino acid sequences were added to the C-terminus of the primary sequence of the nanobody (Figure 1a), including the trimerizing isoleucine zipper domain, as previously described [34]. The VHH-cDNA library generated after the second cycle of immunization was then used for the selection of clones encoding for ai-Nbs (aiChlNP) that specifically bind to the antigen-recognizing site (idiotype) of the aChlNP nanobody but do not bind to the constant nanobody sites. Four obtained ai-Nbs were formatted to make trimerizing Nb derivatives and then were functionally tested in a mouse model. We have shown that the obtained ai-Nbs are able to induce a narrowly specific humoral immune response in the host leading to the generation of intrinsic anti-*Chlamydia* antibodies, potentially therapeutic, specifically recognizing a given antigenic epitope of *Chlamydia*. The immune sera derived from mice immunized with the ai-Nbs were able to suppress chlamydial infection in vitro with slightly varying efficacies. The neutralization results using antibody combinations showed a less than 50% efficacy despite high titers. The most promising ai-Nbs are aiChlNP-74, inducing the highest titer of specific anti-*Chlamydia* IgG in the mice, and aiChlNP-75 and -81, inducing the formation of IgG with the highest protective effect in the chlamydial infection model in vitro. In addition to the analysis of the humoral immune response, we also obtained preliminary data on a model in vitro indicating also the induction of T-cell-mediated immunity as a result of immunization with the obtained ai-Nbs. 

The present study showed that, in response to stimulation with inactivated *C. muridarum*, aiChlNP-74 and aiChlNP-81 are capable of inducing the specific secretion of proinflammatory cytokines such as IFN-γ and TNF-α. In addition, the correlation of the immune response upon stimulation with *C. muridarum* with the immune response upon infection with *C. muridarum* (natural infection) suggests that these ai-Nbs could have therapeutic activity. The T-cells were shown to actively produce TNF-α in response to stimulation with inactivated *C. muridarum*. As with natural infection, we observed a proinflammatory response in the mice to the inactivated *C. muridarum* infection. Thus, TNF-α in combination with IFN-γ may act synergistically to mediate pathogen clearance.

We also observed that IL-6 was actively produced in response to *C. muridarum* infection, but there was no significant production of chlamydial antigens, which excludes the development of pathological changes as a result of immunization with ai-Nbs. At the same time, the production of the anti-inflammatory cytokines IL-4 and IL-10 by the splenocytes was at the same level as in the intact mice. This is possibly due to the fact that high levels of expression of pro-inflammatory cytokines such as IFN-γ, TNF-α and IL-6 can suppress anti-inflammatory cytokines such as IL-4 and IL-10, and vice versa [36,40]. Unlike IFN-γ and TNF-α, IL-2 has no effector function but strongly enhances T-cell expansion [40,41,42]. We have shown that the IL-2 production in response to stimulation with inactivated *C. muridarum* is higher than in response to rTC_0037. But at the same time, we observed that immunization with ai-Nbs, especially aiChlNP-75 and -81, leads to an increase in IL-2 expression to the level seen during natural infection. It can be hypothesized that the balance between IL-2 and pro-/anti-inflammatory cytokines is critical for the appropriate initiation and resolution of the immune response during chlamydial infection. Moreover, such an inflammatory reaction, with increased proliferation of the effector cells due to the production of IL-2, will provide more effective protection after immunization with ai-Nbs in response to infection with *C. muridarum*. Thus, in our study, we showed the immunogenicity of selected ai-Nbs, which could induce a specific immune response due to the balance of the expression of pro-inflammatory and anti-inflammatory cytokines in response to stimulation with chlamydial antigens, further helping neutralize infections caused by *C. muridarum* and its eradication. We suggest further studies on the protective effect of the selected aiChlNP nanobodies be carried out in animal models, such as the already known intravaginal model of chlamydial infection in mice of the DBA/2 line using *C. muridarum* [13].

In this work, we focused our attention on one of the most important virulence factors of *Chlamydia*—the type 3 secretion system (T3SS). Recently, T3SS components have very often been used as candidates for vaccination against other pathogenic microorganisms [8,9,10,11]. The T3SS itself and the injectisome protein TC_0037 (CdsF) are conservative, which is very important when developing vaccine drugs against other bacteria that have the T3SS [13,43]. One of the most important treatments for chlamydial infection, as well as many other intracellular pathogens, is to inhibit the pathogen’s invasion and ability to colonize any host organism. We suggest that the mouse antibodies induced by the ai-Nbs obtained in this work bind the injectisome protein TC_0037 and thereby prevent *Chlamydia* from infecting other cells. Since the target protein in this case is highly conserved, we assume that the obtained ai-Nbs can be effective in the fight against some other types of *Chlamydia* as well. A similar approach can be used to obtain therapeutic nanobodies (ai-Nbs) against T3SS components of other pathogenic microorganisms. In a number of studies, anti-infective nanobodies were obtained that bind to pathogen surface structures, components of secretion and transport systems [44,45,46,47,48]. We hope that the ai-Nb-based method described in this paper could be applied to obtaining biomimetics of “sensitive” pathogenic epitopes of these nanobodies, as well to their use in possible therapeutic directed immunomodulation of the host immune system.

We hypothesize that ai-Nbs and their derivatives (for example, when the ai-Nb coding sequences are delivered into the body and expressed as part of an adenoviral or another vector) may be important specific components of a new subunit vaccine. One of the first pieces of evidence of this possibility was an article back in 2009 about an ai-Nb 1HE isolated from a library generated from a Trastuzumab F(ab’)(2)-immunized llama. 1HE has been shown to closely mimic HER2. Serum from BALB/c mice immunized with 1HE contained anti-HER2 antibodies that inhibited the viability of HER2-positive cells [37]. As a component of a subunit vaccine, ai-Nbs have useful prospects as nanobodies in terms of the very efficient technologies developed for their generation, selection and formatting (modifications and multi-module constructs), for their production and administration, for their safety and for the relatively simplified procedure of their humanization, if required. The lifetime of a nanobody in the bloodstream in case of systemic administration can be adjusted in various ways [23,24,25].

In modern terms, we assume the use of engineered ai-Nbs or their derivatives for the targeted immunomodulation or immunostimulation of the immune cells. Modulating the immune system is a pivotal treatment strategy in modern medicine. Currently, one of the greatest challenges in mAb therapeutics (especially with the systemic use of immunotherapeutic drugs) is their immunogenicity and the formation of anti-drug antibodies which decrease their clinical efficacy [49,50]. In the case of ai-Nbs, it is their immunogenicity that is important for the formation of the host’s own therapeutic antibodies. In addition, systemic administration in this case can potentially be replaced by local (subcutaneous) administration.

The subunit vaccines developed thus far have been found to be poorly immunogenic, and thus multiple boosters and suitable adjuvants are necessary to augment their protective potential [51]. 

## 4. Materials and Methods

### 4.1. Camel Immunization with aChlNP, Construction of a cDNA-VHH Library and the Selection of aiChlNP-Binders

The animal work was approved on 2 February 2018 (registration number 17) by the Commission on Bioethics (formed on 3 May 2017) at the Severtsov Institute of Ecology and Evolution. The camel (*Camelus bactrianus*) used for the immunizations was kept at the Center for Collective Use “Live Collection of Wild Mammals”, and the immunizations were performed at the scientific/experimental base “Chernogolovka” of the Severtsov Institute of Ecology and Evolution at the Russian Academy of Sciences (Chernogolovka, Russia). On day 0 before the immunization, blood was taken for later comparisons of the pre-immune serum with the immune serum. The immunization process consisted of 4 subcutaneous injections (each time, the camel was injected at 4–5 sites in the upper body and neck regions) over a time period of 45 days. In each step, 0.9 mg of (aChlNP-ILZ-HH)x3 was used, and equal volumes of the GERBU adjuvant LQ (GERBU Biotechnik, Heidelberg, Germany) were added to the antigen preparations right before injection. Five days after the final immunization (day 50), 150 mL of blood was taken to check for an aChlNP-specific IgG titer and to isolate the peripheral blood mononuclear cells (PBMCs) to construct a cDNA-VHH-library. An equal volume of phosphate-buffered saline (PBS) containing heparin (100 U/mL) and EDTA (3 mM) was added to the blood to prevent clotting. The aChlNP-specific IgG titer was analyzed using ELISA. Briefly, ELISA plate (MaxiSorp, Nunc, Roskilde, Denmark) wells were coated with monomeric aChlNP-HH (2 µg/mL), blocked with 1% BSA and incubated with immune and pre-immune serum diluted 1:100 to 1:12,800 in 1× PBS. Bound IgG was detected using HRP-conjugated rabbit anti-camel IgG serum (previously obtained by Tillib et al. [34]) and ABTS. The optical density (OD) was measured at 405 nm using a microplate fluorometer.

The preparation of a VHH-cDNA-library and the selection of aChlNP-specific binders using phage display was performed as previously described [33,34]. Nanobodies with different (non-Chlamydial) recognition specificity containing the same conserved framework and amino acid sequences added to their C-terminus as in the case of formatted aChlNPs (e.g., trimerized Nbs [34], available in our laboratory), were used in selection procedures for initial subtraction of nonspecifically bound sticky VHH-phage particles, and then as competitors or blockers of interactions other than anti-idiotype (aiChlNP) with idiotype (aChlNP).

The recombinant protein TC_0037 was used at a high concentration (1 mg/mL) for affinity elution of the VHH phage particles bound to the immobilized aChlNP nanobody at the final stage of the panning procedure. A total of 4 different ai-Nb variants that showed the strongest aChlNP-specific reactivity in ELISA were selected from the original enriched 60 clones. were selected from the original enriched 60 clones.All four selected ai-Nb variants competed with rTC_0037 in binding to the immobilized primary aChlNP nanobody. The selected VHH-DNA sequences (aiChlNP-74, -75, -79 and -81) were subcloned into the expression vector pHEN6 [52] containing the pelB leader sequence for periplasmic expression. Initially selected nanobodies were subjected to formatting. For this purpose, a long 28-amino acid linker (upper hinge region of camel IgG), an additional trimerizing isoleucine zipper (ILZ) domain, HA-tag and His-tag (for detection and purification purposes) were added after the antigen-binding domains at their C-terminus as described previously [34].

### 4.2. Expression and Purification of the Formatted Nanobodies

*E. coli* XL1-blue competent cells (Agilent Technologies, Santa Clara, CA, USA) were transformed with the pHEN6 plasmid encoding the formatted nanobody sequence and grown on LB agar plates with 100 µg/mL ampicillin and 1% glucose. Single colonies were picked and inoculated in 4 mL 2× YT medium with 100 µg/mL ampicillin and 1% glucose at 37 °C. The overnight cultures were transferred into 250 mL of fresh medium (with 60 µg/mL ampicillin and 0.1% glucose) and grown until an OD600 nm of 0.5–0.7 was reached. Expression was induced with 0.2 mM of IPTG overnight at 28 °C. The cultures were centrifuged (3000× *g*, 15 min, 4 °C), and the soluble nanobodies were extracted from the periplasm by resuspending and incubating the pellets with 4 mL of TES buffer (50 mM Tris (pH 8.0), 0.5 mM EDTA, 20% saccharose, 10 mM imidazole, 100 µg/mL PMFS, 5 mM β-mercaptoethanol) for 30 min on ice and subsequently adding 6 mL of Solution 2 (10 mM Tris (pH 8.0), 10 mM imidazole, 1 mM MgCl_2_) for 30 min on ice. The suspension was centrifuged (16,000× *g*, 30 min, 4 °C) and the supernatant purified using HIS-Select^®^ Nickel Affinity Gel (Sigma-Aldrich, St. Louis, MO, USA). The purified nanobodies were dialyzed against 1× PBS containing 10 mM of imidazole and then analyzed using 14% SDS-PAGE under reducing conditions, after which the proteins in the gel were stained with Coomassie Brilliant Blue. 

### 4.3. Reactivity of aChlNP to Recombinant TC_0037

A total of 5 µg/mL of *rTC_0037* and the control protein BSA were coated onto ELISA plates for one hour at 37 °C. The wells were washed with 1× PBS containing 0.05% Tween20 (1× PBST), saturated with 1% BSA in 1× PBST and incubated with the purified formatted nanobodies, monomeric aChlNP-HH or trimerizing aChlNP-ILZ-HH in a series of 4-fold dilutions (from 2.4 to 0.04 µg/mL) in 0.1% BSA/1× PBST for one hour at 37 °C. The wells were washed, and bound nanobodies were detected using a mouse HRP-labeled anti-HA-tag antibody (1:4000 in 0.5% BSA/1× PBST). A color reaction was achieved using the HRP substrate ABTS (1 mg/mL, Sigma-Aldrich). The ODs were measured at 405 nm (with a reference wavelength of 495 nm) using a Tecan Infinite F50 microplate reader (Männedorf, Switzerland) and are shown as means of triplicates ± standard deviation (SD).

### 4.4. Bacteria, Growth Conditions and Cell Lines

*C. muridarum* (Nigg strain) ATCC VR-123 was grown in cycloheximide-treated McCoy cells (a hybrid cell line consisting of human synovial cells and mouse fibroblasts) as described previously [43,53]. The chlamydial elementary bodies (EBs) were collected, purified, quantified and stored at −70 °C in SPG buffer (sucrose/phosphate/glutamic acid: 0.2 mM sucrose, 20 mM sodium phosphate and 5 mM glutamic acid). 

### 4.5. Production of Recombinant Protein TC_0037 TTSS of C. muridarum

As an antigen for selection, we used the recombinant protein TC_0037 produced in *E. coli*, which corresponds to the conservesd protein of the secretion system of the third type (TTSS) in *C. muridarum* [13]. 

### 4.6. Mice

Female mice DBA/2, 8–9 weeks old, free from specific pathogens, were bred and kept in the vivarium of N.F. Gamaleya National Research Center for Epidemiology and Microbiology (Moscow, Russia). The animals were used in accordance with the recommendations of the national guidelines, and the methods of the experiments were approved by the N.F. Gamaleya National Research Center Animal Care Committee.

### 4.7. Immunofluorescence Method for Detecting Intracellular Chlamydia

The eukaryotic cell culture McCoy was infected with *C. trachomatis* or *C. muridarum* as described earlier [43]. A 24 h monolayer of cells was infected with the chlamydia strain by adding a chlamydia culture to the culture medium, followed by centrifugation. The cells were incubated at 37 °C for 48 h and then fixed with acetone and incubated with 5% ovalbumin to prevent non-specific binding. The ability of the nanobodies to bind to chlamydia, which forms intracellular inclusions in eukaryotic cells, was tested using an immunofluorescence assay according to standard protocol [45]. Fixed uninfected cells were used as the control. The detection of bound antibodies was performed using mouse anti-HA antibodies and fluorescein isothiocyanate (FITC)-conjugated secondary antibodies to mouse immunoglobulins.

### 4.8. Immunization of the Mice with Selected ai-Nbs

The DBA/2 mice (6 mice/group) were subcutaneously injected with a series of trimerized ai-Nbs (aiChlNP-74, -75, -79 and -81), either alone or in combination in total volumes of 200 and 400 μL/mouse with Freund’s adjuvant (1:1). The concentrations of the ai-Nbs before they were mixed with the adjuvant were around 1 mg/mL. The immunization was performed three times and four times with an interval of 14 days. The first immunization was performed with complete Freund’s adjuvant (CFA), and subsequent immunizations were performed with incomplete IFA. A total of 14 days after the last immunization, blood was collected from the mice to obtain sera. The blood was centrifuged, and the sera were obtained and stored at −20 °C. To assess the immunogenicity of the aiChlNP nanobodies, an enzyme-linked immunosorbent assay (ELISA) was performed to detect antibodies specific to the rTC_0037 protein and to analyze the neutralizing activity of the obtained antibodies. Pre-immune sera obtained from intact mice were used as negative controls. 

To evaluate specific antibodies of IgG isotypes (IgG1 and IgG2a) and the cellular immune response, immunization with the most immunogenic ai-Nbs (aiChlNP-74, -75 and -81) was performed. Four stages of immunization were carried out with an interval of 2 weeks. The anti-idiotypic nanobodies aiChlNP-74, -75 and -81 were injected subcutaneously into the DBA/2 mice in combination with Freund’s adjuvant (as described previously), while the last immunization was performed using FAMA (GERBU Biotechnik) as the adjuvant to achieve the best effect using small volumes when administered to animals. Intact mice injected with PBS with Freund’s and FAMA adjuvants were used as the negative controls. For the evaluation of the IgG1- and IgG2a-specific antibodies, mouse sera were obtained after 4-fold immunization. Mouse spleens were used to evaluate the IFN-γ-specific T-cell immune response after immunization. The spleen cells (2 × 10^5^ cells/mL) were incubated with UV-inactivated *C. muridarum* and rTC_0037 for 24 h in complete RPMI-1640 medium (Gibco, Carlsbad, CA, USA) containing 5% fetal bovine serum, 2 mM of L-glutamine and 1% penicillin-streptomycin (Gibco, Carlsbad, CA, USA). Spleens from mice infected intravaginally with 10^4^ IFU/mL *C. muridarum* were used as the positive controls. Spleens from mice immunized with adjuvants and from the intact mice were used as negative controls.

### 4.9. Indirect Enzyme-Linked Immunosorbent Assay (ELISA) Method

The ELISA method was used to determine the specific IgG antibodies as well as the IgG1 and IgG2a isotypes in the serum of the mice. For the assay, 96-well microtiter ELISA plates (Nunc, Rochester, NY, USA) were used. The rTC_0037 protein was diluted to a concentration of 10 μg/mL in PBS, applied in a volume of 100 μL into the wells of the plates, after which an incubation was carried out overnight at 4 °C. After incubation, the plates were washed several times with PBS, and non-specific binding was blocked using 0.1% bovine serum albumin (BSA) in PBS for 30 min at room temperature. The mouse sera obtained after immunization were titrated using 0.1% BSA in PBS for dilution and then incubated overnight at 4 °C. The plates were then washed 3 times with 0.1% BSA in PBS. Biotinylated anti-mouse IgG monoclonal antibodies (IgG1 and IgG2a) (BioLegend, San Diego, CA, USA) used at a dilution of 1:1000 were added to the wells of the plates and incubated for 1 h at room temperature (20 °C), after which the streptavidin HRP dye (BioLegend, San Diego, CA, USA) and TMB substrate (BioLegend, San Diego, CA, USA) were added. The reaction was stopped after 20 min according to the addition of 50 μL/well 1 M H_2_SO_4_. The absorbance at 450 nm was determined using a Multiskan EX microplate reader (Thermo Fisher Scientific Oy, Vantaa, Finland).

### 4.10. In Vitro Neutralization Assay

The neutralization assays of *C. muridarum* were performed in the McCoy cell culture in 24-well culture plates with glass slides (12 mm diameter), as described in [13]. The immune mice sera were diluted in Dulbecco’s modified Eagle medium, DMEM (PanEco, Moscow, Russia). To 100 μL of the serial dilutions of the sera (1:32–1:528), 100 μL of the *C. muridarum* EB suspension in DMEM medium (1.4 × 10^5^ IFU/mL) was added for 30 min at 37 °C on a shaker at 150 rpm. The samples incubated with EB without sera were used as the infection controls. After incubation, suspensions of 100 μL/well (in duplicate) diluted with 900 μL of DMEM containing 10% fetal bovine serum and 1 mg/mL cycloheximide were added to McCoy cells in 24-well culture plates.

The plates were centrifuged at 800× *g* for 1 h and then incubated at 37 °C for 48 h. Next, the cells adsorbed onto the glass were fixed with ethanol and stained with monoclonal antibodies conjugated with fluorescein isothiocyanate (FITC) against *Chlamydia* lipopolysaccharides (Nearmedic Plus, Moscow, Russia). The cells containing inclusions were analyzed using a Nikon Eclipse 50i fluorescence microscope (Nikon, Amsterdam, The Netherlands) at 100× magnification and counted to determine the percentage of infected cells in the McCoy monolayer.

### 4.11. ELISPOT Assay

The *C. muridarum*- and TC_0037-specific cytokine-producing T-cell immune response was assessed using ELISPOT. The assay was performed using a commercial IFN-γ ELISPOT Ready-SET-Go! kit (eBioscience, Inc., San Diego, CA, USA). For the assay, 96-well MultiScreen-IP filter plates (Millipore, Billerica, MA, USA) with immobilized IFN-γ antibodies plated onto them were used. The plates were incubated overnight at 4 °C. After incubation, the plates were washed with sterile PBS. T-cells were isolated from the mouse spleens using T-cell separation columns (Cederlane, Burlington, ON, Canada) according to the manufacturer’s protocols. Enriched cell suspensions were added to the plates at a concentration of 2 × 10^5^ cells per well (in 100 μL of DMEM culture medium) in the presence of the UV-treated *C. muridarum* EB at a final concentration of 10^4^ IFU/mL and rTC_0037 at a final concentration of 10 μg/mL. The plates were incubated for 24 h at 4 °C. Next, washing with PBS and incubation with the biotinylated anti-IFN-γ monoclonal antibodies was performed. Stains (spots) were visualized using Avidin HRP dye and a freshly prepared 3-amino-9-ethylcarbazole (AEC) substrate solution. The spots were quantitatively counted using the AID EliSpot Reader (Autoimmun Diagnostika GmbH, Strassberg, Germany). ELISPOT was carried out similarly using kits for TNFα, IL-6 and the inflammatory cytokines IL-4, IL-10 and IL-2 (eBioscience, Inc., San Diego, CA, USA). 

## 5. Conclusions

The proposed method for generating and using ai-Nbs, which imitate and present to the host’s immune system exactly the desired region of an antigen, creates a fundamentally new universal approach to the creation of molecular antigen-mimicking structures that can be useful both for immunoassays and as part of specific vaccines for targeted immunostimulation, especially useful in cases where it is difficult to obtain an antigen that retains the desired epitope in its native conformation.

## Figures and Tables

**Figure 1 ijms-25-02047-f001:**
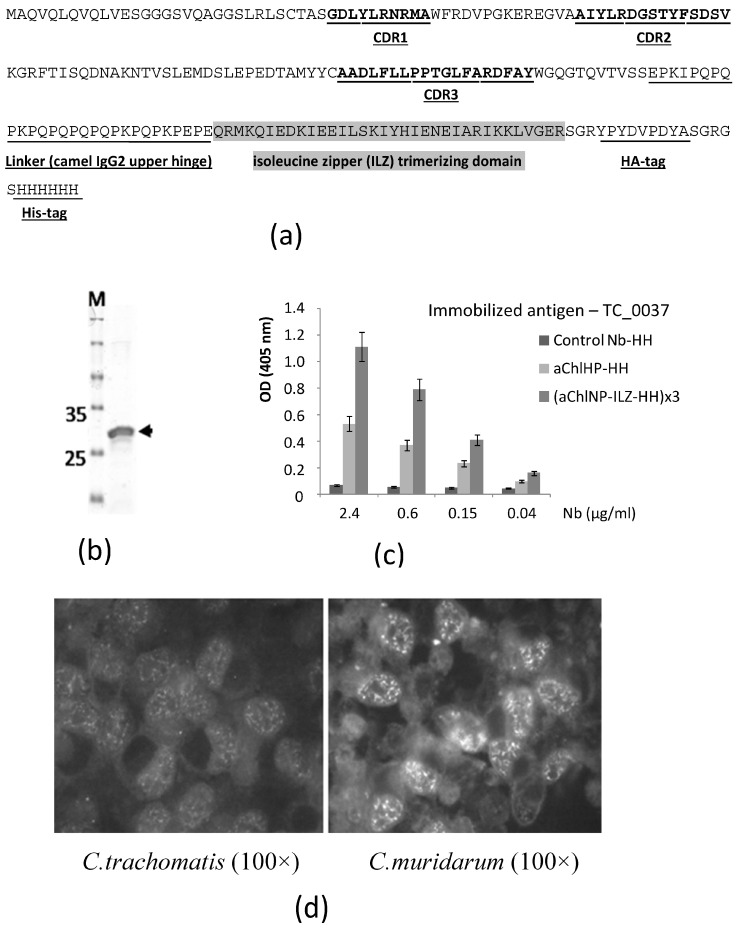
Characteristics of nanobody aChlNP recognizing Chlamydial Type III Secretion System Needle Protein. (**a**) Amino acid sequence of aChlNP. Complementarity-determining regions (CDRs, underlined and in bold) 1–3 are indicated. The linker sequence (underlined), isoleucine zipper (ILZ)-trimerizing domain (marked in gray) and tag sequences (HA-tag and His-tag, underlined) are added to the C-terminus of the initially selected sequence (VHH) to make its trimerizing modification, aChlNP-ILZ-HH. (**b**) Expression and purification of aChlNP-ILZ-HH verified using SDS-PAGE under reducing conditions (the arrow indicates the position of the protein monomer). (**c**) Reactivity of aChlNP in two formats and unrelated control nanobody in serial dilutions to plate-bind the TC_0037 protein of *C. muridarum*. OD values (*y*-axis) correspond to the amount of bound nanobodies and are shown as means of triplicates ± SD. (**d**) The trimerizing nanobody (aChlNP-ILZ-HH)x3 binds to *C. trachomatis* or *C. muridarum*, forming intracellular inclusions in infected eukaryotic McCoy cells. Images were observed at 100× magnification.

**Figure 2 ijms-25-02047-f002:**
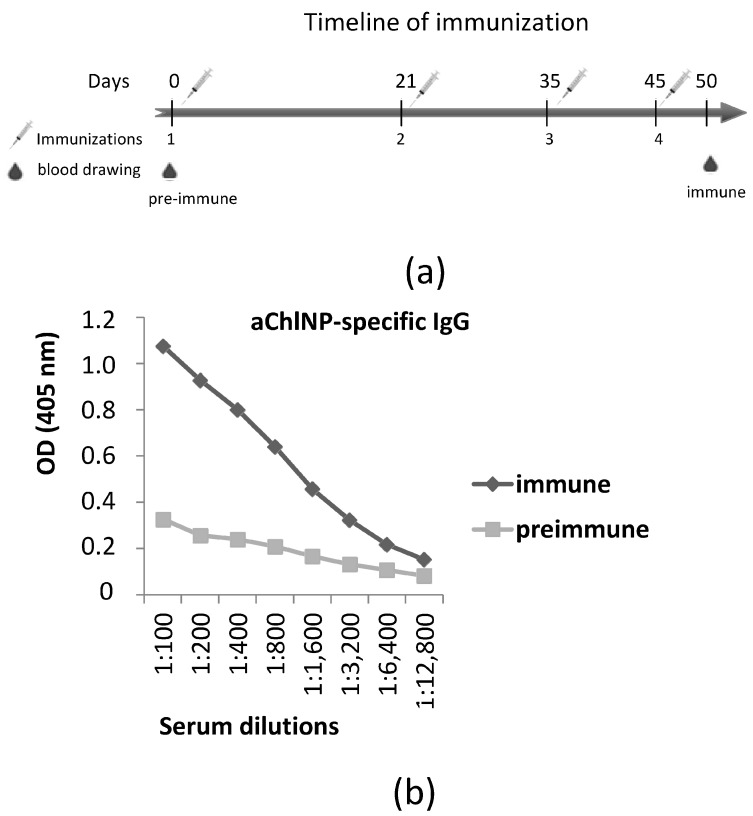
Generation of anti-aChlNP nanobodies (aiChlNP nanobodies). (**a**) shows the timeline of immunization. Intervals of injections with (aChlNP-ILZ)x3 as well as of obtaining pre-immune and immune sera are indicated (in days). The time point of isolating peripheral blood mononuclear cells (PBMCs) for generating a VHH-cDNA library is marked. (**b**) Reactivity of IgG antibodies specific to aChlNP in pre-immune serum (taken on day 0) and immune serum (taken on day 50 after four injections with aChlNP-ILZ) was determined using ELISA. OD values (*y*-axis) correspond to the amount of bound IgG antibodies and are shown as mean of duplicates with a variation of less than 10%. Displayed data are representative of two independent experiments.

**Figure 3 ijms-25-02047-f003:**
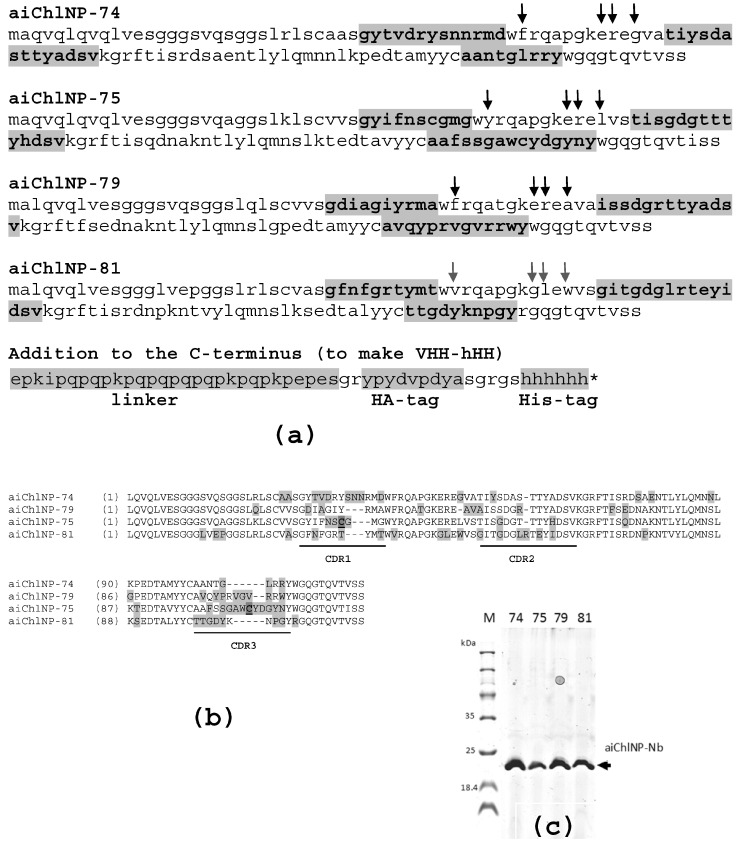
Characteristics of four selected anti-idiotypic nanobodies (aiChlNP-74, aiChlNP-75, aiChlNP-79 and aiChlNP-81). (**a**) Amino acid sequences are presented. Complementarity-determining regions (CDR, highlighted in bold and gray) 1–3 are indicated. Positions of hallmark VHH amino acid substitutions (V37F, G44E, L45R, W47G) are marked with arrows. Below is the additional sequence added to the C-terminus of all primary nanobodies during formatting. An asterisk (*) indicates a stop codon. (**b**) Comparison of amino acid sequences of four selected variants of aiChlNPs (using Vector NTI 10.0.1 software). Non-similar amino acids are highlighted in gray; additional cysteins in CDRs are bold and underlined. (**c**) Expression and purification of aiChlNP-HH nanobodies is checked using SDS-PAGE under non-reducing conditions. The expected size of these proteins is about 20 kDa.

**Figure 4 ijms-25-02047-f004:**
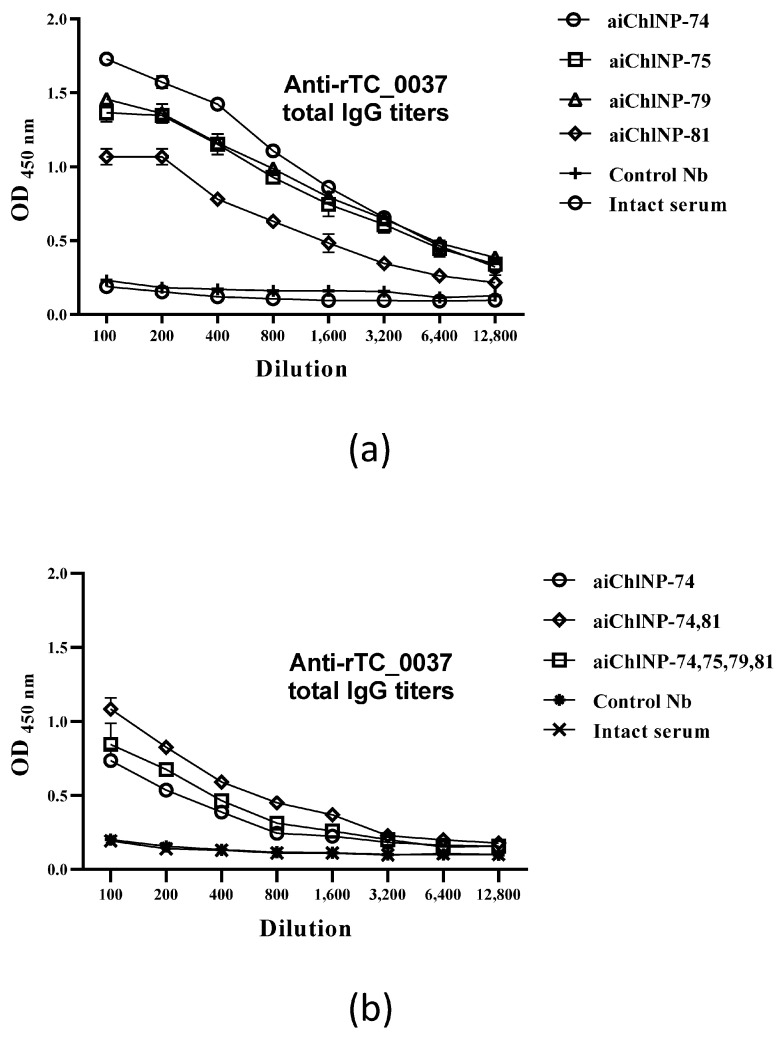
Specific IgG antibody responses to TC_0037 following immunization with ai-Nbs. (**a**) ai-Nbs (aiChlNP-74, -75, -79, -81) used for four-fold immunization of mice. (**b**) IgG titers upon three-fold immunization with aiChlNP-74 in combination with other ai-Nbs. Control Nbs (trimerized Nbs of different specificity) and intact serum are negative controls. TC_0037-specific IgG titers were determined using enzyme-linked immunosorbent assay (ELISA) and expressed as the mean ± SEM of groups of six mice from two experiments (*p* < 0.05).

**Figure 5 ijms-25-02047-f005:**
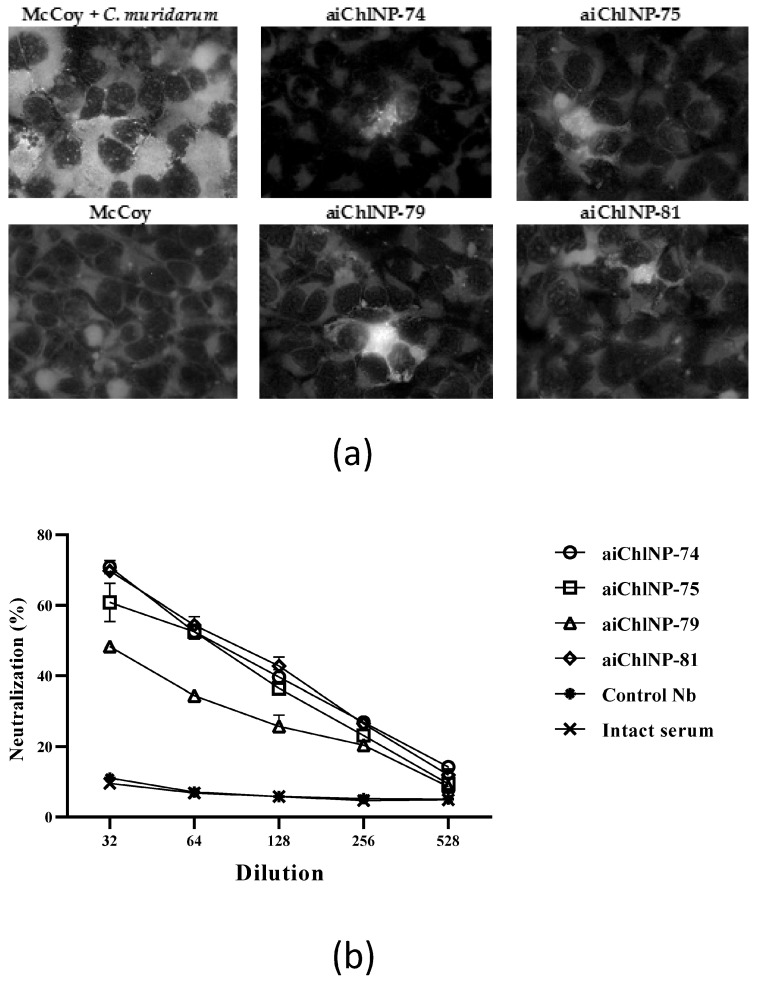
Evaluation of the neutralizing activity of the obtained serum antibodies in response to ai-Nb (aiChlNP-74, -75, -79 and -81) immunization. (**a**) The percentage of neutralizing activity of antibodies against *C. muridarum* infection in the McCoy cell line was evaluated. Images were observed at 100× magnification. (**b**) The number of chlamydial inclusions was counted 48 h after cells were infected with 1.4 × 10^5^ IFU/mL *C. muridarum* culture with pre-immune and immune sera. Anti-control Nb immune and intact sera were negative controls. Neutralizing activity was determined by measuring the reduction in the number of inclusions formed by antibody-opsonized elementary cells. Graphs represent the percentage of neutralization as the mean of experiments performed with three repetitions, with standard deviations (*p* < 0.01).

**Figure 6 ijms-25-02047-f006:**
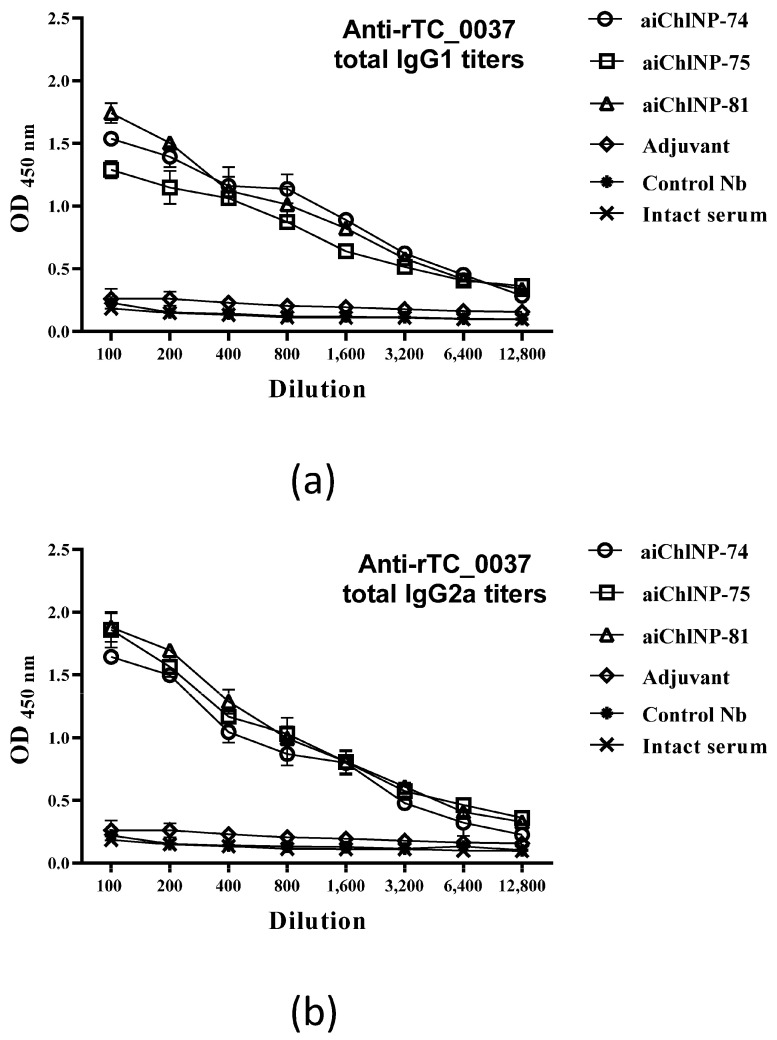
Determination of IgG1 and IgG2a isotypes in sera of mice immunized with aiChlNP Nbs. (**a**) Titers of specific IgG1 antibodies. (**b**) Titers of specific IgG2a antibodies. Titers of IgG isotype specific to rTC_0037 were determined using solid phase enzyme-linked immunosorbent assay (ELISA) and expressed as the mean ± standard error of the mean for groups of six mice from two experiments (*p* < 0.05).

**Figure 7 ijms-25-02047-f007:**
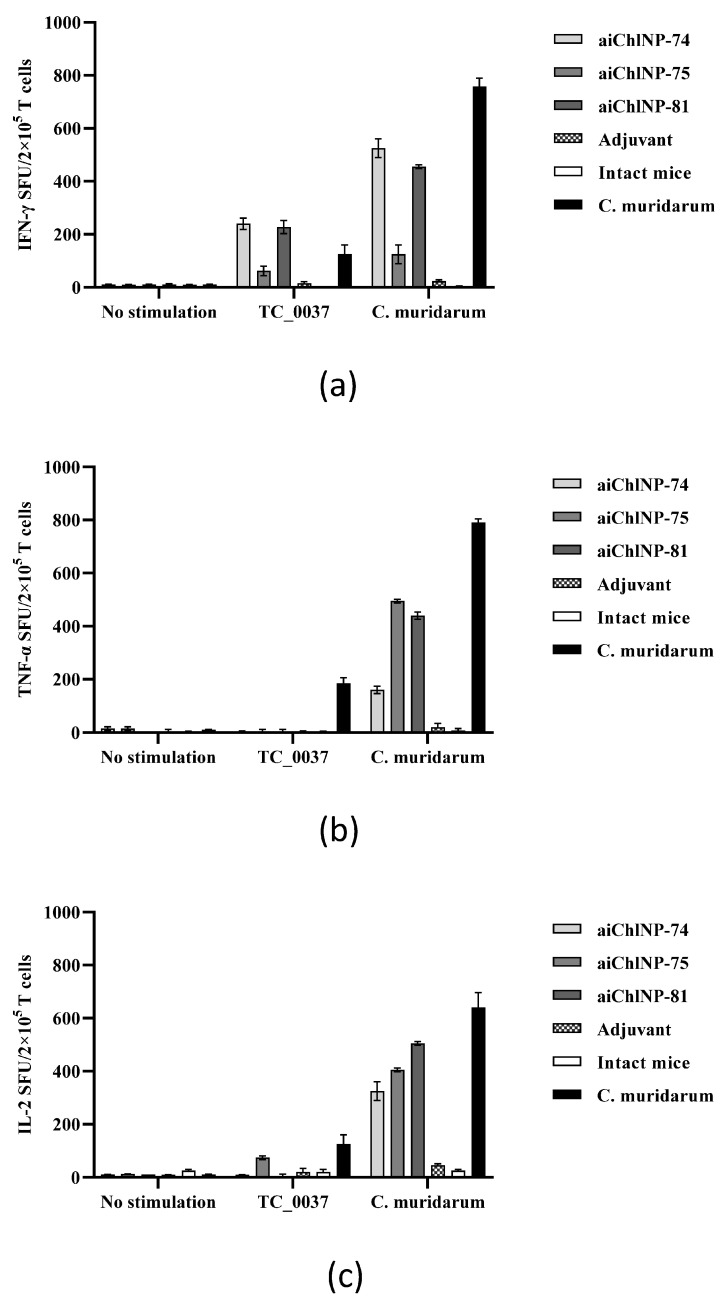
aiChlNP immunization induces in mice secretion by the T-cells of (**a**) IFN-γ, (**b**) TNF-α, (**c**) IL-2. DBA/2 mice (6 mice/group) were immunized subcutaneously with ai-Nbs in combination with Freund’s and FAMA adjuvants. Mice intravaginally infected with *C. muridarum* were used as controls. Two weeks after the last immunization, T-cells were isolated from splenocytes and stimulated in vitro with rTC_0037 or UV-inactivated *C. muridarum*. ELISPOT (stain-forming units [POEs]) in response to rTC_0037 and *C. muridarum* (UV) stimulation was quantitatively analyzed. Results are presented as mean POEs per 2 × 10^5^ splenocytes ± SEM. Adjuvant-immunized and naïve mice were negative controls.

## Data Availability

The nucleotide sequences of the nanobodies obtained and characterized in this study (and the protein sequences deduced from them) have been submitted to the GenBank database under accession numbers OR885932, OR901958, OR901959, OR901960 and OR901961.

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
