# Peer review of "Anti-Idiotypic Nanobodies Mimicking an Epitope of the Needle Protein of the Chlamydial Type III Secretion System for Targeted Immune Stimulation"

_ijms, 2024, doi:10.3390/ijms25042047_

Round 1

Reviewer 1 Report

Comments and Suggestions for Authors

The authors tried to produce anti-chlamydia molecules and for the subsequent obtaining of anti-idiotypic nanobodies (ai-Nbs) 19 mimicking the structure of a given epitope of the pathogen. The topic is interesting, however, the nanobodies were not successfully produced. And the hypothesis was not well proved. 

1.      P4, Line 165, how to prove ChlNP is with trimerizing ILZ-domain?

2.      P9, Line 257-259, the binding affinity of the produced antibodies is very low. The absorbance values at the titer of 1:12800 are very close to the background noise.

3.      P9, Line 266, title 2.4 need to be revised.

4.      P11, Line 294, how to calculate the ratio of specific antibodies IgG2a/IgG1?

5.      P17, Line 540, the title of 4.8.2 is similar to 4.8.1 which makes audience confused.

6.    P17, Line 544, the title of 4.8.2 said it was immunized with antigens, but here it was injected with antibodies.

Comments on the Quality of English Language

The English needs to be well polished.

Author Response

Thank you very much for taking the time to review this manuscript!

Please find the detailed responses below and the corresponding corrections and additions in the text of the article highlighted in yellow. Figures 4 and 5 were also corrected.

«The authors tried to produce anti-chlamydia molecules and for the subsequent obtaining of anti-idiotypic nanobodies (ai-Nbs) 19 mimicking the structure of a given epitope of the pathogen. The topic is interesting, however, the nanobodies were not successfully produced. And the hypothesis was not well proved.»

- We cannot agree with this opinion and assume that the reviewer did not quite understand the essence of this quite pioneering study. This may be due to our insufficiently clear presentation of the main idea and main results when writing the article, and we agree with the reviewers that the article can certainly be improved. We tried to correct the text of the article and Figures 4 and 5 taking into account all the comments made by the reviewers. Changes to the text are highlighted in yellow.

  1. P4, Line 165, how to prove ChlNP is with trimerizing ILZ-domain?

- The preparation of trimerized nanobody derivatives has long been a routine procedure in our laboratory. This procedure was first described in detail by us 10 years ago in article [34] mentioned just above in the text on line 159.

Оn Fig. 1b, it can be seen that the monomer size of the formatted nanobody with the trimerizing domain  corresponds to approximately 27 kDa, which is in good agreement with the expected (calculated based on the amino acid sequence indicated in Fig. 1a) size of such a formatted nanobody. We have added additional evidence to the text near this point:

Before analysis by SDS-PAAG (Figure 1, b), the trimerized nanobody ChlNP was additionally purified from small impurities and, presumably, from monomeric nanobodies by ultrafiltration on a filter with MWCO 50 kDa (Vivaspin Turbo 4, Sartorius). This nanobody remained almost entirely at the top and did not pass through this filter.

  1. P9, Line 257-259, the binding affinity of the produced antibodies is very low. The absorbance values at the titer of 1:12800 are very close to the background noise.

- The reviewer may not have fully understood that in this case, the antigens used for mouse immunization were anti-idiotypic "camel" nanobodies (ai-Nb) selected to the primary "camel" nanobody (recognizing the Chlamydia target protein), and the increase in mouse antibody titer after immunization with ai-Nb was monitored against the recombinant Chlamydia target protein. What is fundamentally important here is not so much the level of induction as the fact itself, and this is the main thing in this pioneering article, that the selected (following the camel immunization) anti-idiotypic nanobodies cause a given specific immune response in mice against patogen.

For better clarity, we have added additional information to the text of the article:

In the case of immunization with primary aChlNP or trimerized nanobodies of a different specificity (nanobodies against the hemagglutinin of the influenza A virus [34] as a control, Control Nb ), we, similarly to the control of intact serum, did not observe any increase in the titer of mouse antibodies binding to the chlamydial antigen ТС_0037.

We have adjusted Figures 4 and 5 to include data from these controls (with Control Nb).

Please look at th fig. 4 (a). We see here a clear specific anti-TC_0037 andibody titre growth in immunized mice only in the case of injections of selected ai-Nbs, and not random nanobodies with the same formatting. Signals in ELISA may vary from trial to trial, but the increase in titer of anti-Chlamydia-specific mouse antibodies compared to negative controls was clear and reproducible for all ai-Nbs obtained in this study.

It is clear that the observed increase in the titer of specific mouse antibodies can be significantly increased by using special formatting of ai-NT, carrier proteins and more effective adjuvants. But this was not the main focus of this study.

  1. P9, Line 266, title 2.4 need to be revised.

- We revised the title to the following:

Immune sera derived from mice immunized with aiNbs are able to partly suppress chlamydial infection in a neutralization assay in vitro.

  1. P11, Line 294, how to calculate the ratio of specific antibodies IgG2a/IgG1?

- We corrected the text:

     The IgG2a/IgG1 ratio of specific anti-TC_0037 antibodies, determined in sera diluted 1:100 as             the average of six OD450 (IgG2a)/OD450 (IgG1) measurements (+SE), was greater than 1 in    case of all three aiNbs, which may indicate a differentiated immune response predominantly to   type 1 T helper cells (Figure 6).

This calculation method was used previously, for example in the work [Sukumar Pal, Ida Theodor, Ellena M. Peterson, and Luis M. de la Maza*. Immunization with the Chlamydia trachomatis Mouse Pneumonitis Major Outer Membrane Protein Can Elicit a Protective Immune Response against a Genital Challenge. Infect Immun. 2001 Oct; 69(10): 6240–6247.]

  1. P17, Line 540, the title of 4.8.2 is similar to 4.8.1 which makes audience confused.

- We have combined these two subsections into one with one common title.

  1. P17, Line 544, the title of 4.8.2 said it was immunized with antigens, but here it was injected with antibodies.

-  It was corrected.

Thank you.

Reviewer 2 Report

Comments and Suggestions for Authors

The research is interesting. However, the authors should address the below made major comments before it can be accepted for publication.

1.     It would be beneficial to mention how critical it is to identify the target antigens for nanobodies. The inability to identify specific antigens in the earlier stages is a common challenge, and it would be interesting to discuss the implications of this and how the use of TC_0037 as an antigen addressed this issue.

2.     Elaborate on why TC_0037 was chosen as the antigen for refined selection procedures. What is its relevance to Chlamydial Type III Secretion System Needle Protein, and how does it contribute to the study's objectives?

3.     Clarify the criteria used for the selection of the four aiNbs (aiChlNP-74, aiChlNP-75, aiChlNP-79, and aiChlNP-81). What characteristics were desirable in these selected antibodies?

4.     Discuss the significance of the study's findings regarding the ability of ai-Nbs to induce the formation of specific IgG antibodies in mice. How might this contribute to potential therapeutic applications?

5.     Explain the rationale for combining ai-Nbs in the immunization process. What advantages or synergistic effects are expected from the combination compared to using individual ai-Nbs?

6.     Consider discussing the balance between pro-inflammatory cytokines (such as IFN-γ, TNF-α, and IL-6) and anti-inflammatory cytokines (such as IL-4 and IL-10) and how this balance might contribute to an effective immune response.

7.     Discuss how the method developed in this study compares to existing methods for generating nanobodies or anti-idiotypic nanobodies. Highlight any unique features or advantages of the developed approach.

8.     Expand on the potential therapeutic applications of the developed aiNbs. Consider discussing how these antibodies might be used in the context of treating Chlamydia infections and whether they could be applied to other infectious diseases.

9.     Consider discussing how the use of aiNbs as a component of a subunit vaccine compares to other vaccine approaches, especially in terms of efficacy, safety, and ease of administration.

10.  Discuss the potential impact of aiNbs on the field of monoclonal antibody (mAb) therapeutics. Consider how the immunogenicity of aiNbs may differ from traditional mAbs and how this might influence their clinical utility.

Author Response

Thank you very much for taking the time to review this manuscript! Please find the detailed responses below and the corresponding revisions/corrections in the article text highlighted in yellow in the attaced file. Figures 4 and 5 were also corrected.

'The research is interesting. However, the authors should address the below made major comments before it can be accepted for publication.'

- Thank you for your opinion and valuable comments!

  1. It would be beneficial to mention how critical it is to identify the target antigens for nanobodies. The inability to identify specific antigens in the earlier stages is a common challenge, and it would be interesting to discuss the implications of this and how the use of TC_0037 as an antigen addressed this issue.

- We agree with this and tried to better highlight this important point in the introduction of the article:

The type 3 secretion system (T3SS) is the predominant virulence factor of chlamydiae. It is essential for cell invasion and is active at all life stages [5,6]. Some T3SS proteins are on the surface and can be targeted by neutralizing antibodies. A T-cell response to T3SS antigens has recently been shown to be associated with protection against trachomatis infection in highly exposed women [7], and T3SS components have recently attracted attention as vaccine candidates against other pathogenic bacteria [8-11]. The C. trachomatis T3SS filament protein, CdsF, and its orthologs in other bacteria such as TC_0037 protein of Chlamydia muridarum form the needles of injectisomes and are believed to facilitate the insertion of translocators into the host cell membrane [5,6,12]. CdsF (TC_0037) is highly conserved, showing 95% sequence identity in the genus Chlamydia. It is abundant on bacterial surfaces, raising the possibility that a CdsF (TC_0037)-based vaccine may induce a wide range of protection against all medically significant strains. It was shown that the T3SS needle protein TC_0037 induced specific humoral and T cell responses, decreased Chlamydia loads in the genital tract, and abrogated pathology of upper genital organs. It could be a good candidate for inclusion in a Chlamydia vaccine [13]. In the study just mentioned, the authors immunized mice with a replication-defective adenoviral vector expressing the recombinant target antigen TC_0037. Thus, TC_0037, based on published data, is a very promising target for immunotherapeutic suppression of chlamydial infection. The presented work was made possible due to the fact that at the previous stage of research we were able to obtain highly specific single-domain antibodies (nanobodies) that recognize precisely this promising therapeutic target, TC_0037.

as well as to mention it in the discussion section:

For the described method, it is fundamentally important to identify a promising therapeutic target at the beginning of the work and to be able to obtain primary single-domain antibodies that highly specifically recognize it and have therapeutic potential. At the beginning of the presented work we were able to obtain the desired highly specific nanobody that recognizes precisely such a promising therapeutic target, Chlamydial Type III Secretion System Needle Protein TC_0037.

  1. Elaborate on why TC_0037 was chosen as the antigen for refined selection procedures. What is its relevance to Chlamydial Type III Secretion System Needle Protein, and how does it contribute to the study's objectives?

- We already highlighted these questions above in response to the first comment.

  1. Clarify the criteria used for the selection of the four aiNbs (aiChlNP-74, aiChlNP-75, aiChlNP-79, and aiChlNP-81). What characteristics were desirable in these selected antibodies?

 - To clarify the criteria for the selection of aiNbs, we have supplemented the text in the Materials and Methods section:

Nanobodies of different recognition specificity having the same conservative (framework) and added amino acid sequences to the C-terminus as formatted aChlNP (trimerized Nbs [34] available in our lab) were used for the preliminary subtraction of VHH-phage particles and then as competitors or blockers of interactions other than anti-idiotype (aiChlNP) with idiotype (aChlNP). The recombinant protein TC_0037 was used at a high concentration (1 mg/ml) for affinity elution of VHH-phage particles bound to the immobilized aChlNP nanobody at the final stage of the panning procedure. Four different aiNb variants that showed the strongest aChlNP-specific reactivity in ELISA were selected from 60 finally enriched clones. All four selected aiNb variants competed with rTC_0037 for binding to the immobilized primary aChlNP nanobody.

and also further formulated the main criteria in the Discussion section:

The main desirable criteria for the selection of aiNb clones at this stage are: a) high specificity and affinity of the binding of selected aiNb variants to a unique idiotype (paratope), and not to conservative regions of the primary nanobody, and b) aiNb should effectively compete with the original target antigen (in this case - TC_0037) for binding to the primary nanobody.

  1. Discuss the significance of the study's findings regarding the ability of ai-Nbs to induce the formation of specific IgG antibodies in mice. How might this contribute to potential therapeutic applications?

- In the Abstract of the article we indicated that

We hypothesize that the proposed method of creation and the use of ai-Nb, which mimics and presents to the host immune system exactly the desired region of the antigen, creates a fundamentally new universal approach to generate molecular structures as a part of specific vaccine for targeted induction of immune response, especially useful in cases where it is difficult to prepare an antigen preserving the desired epitope in native conformation.

We have added additional text to the discussion section:

According to our preliminary data, the method for obtaining ai-Nbs described in this article is quite universal. We are extremely encouraged by the results obtained with the chlamydial target (as well as with several other targets in our laboratory) and see great scientific and practical potential in using the described approach to produce biomimetics of the epitopes of many biomedically important target antigens. Such biomimetics can be used to replace certain toxic antigens in diagnostic methods, as already demonstrated [27-29], and formulate subunit vaccines to specifically induce a host immune response against a specific therapeutic target epitope.

  1. Explain the rationale for combining ai-Nbs in the immunization process. What advantages or synergistic effects are expected from the combination compared to using individual ai-Nbs?

- We added a corresponding explanation in the Results section 2.3:

Since we do not yet know the features of both the dynamic nature of the recognizable epitope and the characteristics (for example, the degree of stabilization of paratope loops) of specific selected ai-Nbs, one could assume the possibility of a synergistic effect of using a combination of the obtained ai-Nbs to induce the corresponding antibodies in the host body with small features of their paratopes that could theoretically bind the target pathogenic epitope more accurately in slightly different dynamics.This work did not reveal a significant synergistic effect.

  1. Consider discussing the balance between pro-inflammatory cytokines (such as IFN-γ, TNF-α, and IL-6) and anti-inflammatory cytokines (such as IL-4 and IL-10) and how this balance might contribute to an effective immune response.

- We have added the following text to the Discussion section.

 The present study showed that, in response to stimulation of inactivated C. muridarum, aiChlNP-74 and aiChlNP-81 are capable of inducing the specific secretion of proinflammatory cytokines such as IFN-γ and TNF-α. In addition, the correlation of the immune response upon stimulation with C. muridarum with the immune response upon infection with C. muridarum (natural infection) suggests that these aiNbs could have a therapeutic activity. T cells were shown to actively produce TNF-α in response to stimulation with inactivated C. muridarum. As with natural infection, we observed a proinflammatory response in mice to C. muridarum infection. Thus, TNF-α in combination with IFN-γ may act synergistically to mediate pathogen clearance.

We also observed that IL-6 was actively produced in response to C. muridarum infection, but there was no significant production of chlamydial antigen, which excludes the development of pathological changes as a result of immunization with iaNbs. At the same time, the production of anti-inflammatory cytokines IL-4 and IL-10 by splenocytes was at the level of intact mice. This is possibly due to the fact that high levels of expression of pro-inflammatory cytokines such as IFN-γ, TNF-α and IL-6 can suppress anti-inflammatory cytokines such as IL-4 and IL-10 and vice versa [36, 42]. Unlike IFN-γ and TNF-α, IL-2 has no effector function but strongly enhances T cell expansion [42, 43, 44]. We have shown that IL-2 production in response to stimulation with inactivated C. muridarum is higher than in response to rTC_0037. But at the same time, we observe that immunization with ai-Nbs, especially aiChlNP-75 and -81, leads to an increase in IL-2 expression to the level during natural infection. It can be hypothesized that the balance between IL-2 and pro-, anti-inflammatory cytokines is critical for the appropriate initiation and resolution of immune responses during chlamydial infection. Moreover, such an inflammatory reaction with increased proliferation of effector cells due to the production of IL-2 will provide more effective protection after immunization with aiNbs in response to infection with C. muridarum infection.Thus, in our study, we showed the immunogenicity of selected ai-Nbs that could induce a specific immune response due to the balance of expression of pro-inflammatory and anti-inflammatory cytokines in response to stimulation with chlamydial antigens, which could further help neutralize infection caused by C. muridarum and its eradication.

  1. Discuss how the method developed in this study compares to existing methods for generating nanobodies or anti-idiotypic nanobodies. Highlight any unique features or advantages of the developed approach.

- We have added the following text to the Discussion section

The main proposed stages for obtaining ai-nanobodies are as follows.

1) For the described method, it is fundamentally important to identify a promising therapeutic target at the beginning of the work and to be able to obtain primary single-domain antibodies that highly specifically recognize it and have therapeutic potential. Most likely, this will be an already known target for which monoclonal antibodies (mAb) have been obtained and the immunotherapeutic effect of using these antibodies has been demonstrated.

2) Generation of primary nanobodies against a selected target. Here we have a classic case of obtaining new nanobodies using a well-established technological platform. From the panel of obtained nanobodies, it is necessary to functionally select those variants that effectively bind to the selected target in vivo and have the desired effect (for example, they can suppress the infectivity of Chlamydia in vitro).

3) The functionally selected primary nanobody must be formatted in such a way as to most effectively use it as an antigen for subsequent immunization of a member of the Camelidae family (using special formatting, carrier proteins and effective adjuvants). From our experience, a parallel trimerized version of the nanobody reproducibly gave a good result in inducing the formation of ai-Nb precursors when immunizing a camel. The use of special formatting to obtain trimerizing nanobodies is a development of our laboratory [34].

Among some other published examples of the use of anti-idiotypic nanobodies, which are still rare but show great potential, nanobodies usually produced against idiotypes of classical antibodies [26-29, 37-41]. We consider this as an alternative possible route, however, in our experience, the efficiency of generation and selection of ai-Nbs that mimic the epitope of the original target increases significantly if nanobodies are used as primary idiotypes instead of mAbs.

4) Generation of ai-Nbs using a well-established technological Nanobody platform. The main desirable criteria for the selection of aiNb clones at this stage are: a) high specificity and affinity of the binding of selected aiNb variants to a unique idiotype (paratope), and not to conservative regions of the primary nanobody, and b) aiNb should effectively compete with the original target antigen (in this case - TC_0037) for binding to the primary nanobody.

5) Selected ai-Nbs must be formatted in such a way as to most effectively use it as an antigen for subsequent immunization of a model organism (mouse). For the initial demonstration experiments (in this article), we also used a trimerized version of ai-Nb, but this is only one of the possible formats for preparing antigenic material.

6) Immunization of model animal (mouse) with an antigenic material based on formatted ai-Nb (or its multi-module derivatives)  and functional analysis of induced host antibodies. The goal is to select ai-Nb that is able to induce a narrowly specific humoral immune response of the host leading to generation  of intristic antibodies against initially selected therapeutic target.

  1. Expand on the potential therapeutic applications of the developed aiNbs. Consider discussing how these antibodies might be used in the context of treating Chlamydia infections and whether they could be applied to other infectious diseases.

- We have added the following text to the Discussion section.

In this work, we focused our attention on one of the most important virulence factors of Chlamydia - The type 3 secretion system (T3SS). T3SS components have recently been very often used as candidates for vaccination against other pathogenic microorganisms [8-11]. The T3SS itself and the injectisome protein TC_0037 (CdsF) are conservative, which is very important when developing vaccine drugs against other bacteria that have the T3SS [13, 45]. One of the most important treatments for chlamydial infection, as well as many other intracellular pathogens, is to inhibit the pathogen's invasion and ability to colonize any host organism. We suggest that mouse antibodies induced by the ai-Nbs obtained in this work bind the injectisome protein TC_0037 and thereby prevent Chlamydia from infecting other cells. Since the target protein in this case is highly conserved, we assume that the obtained aiNbs can be effective in the fight against some other types of Chlamydia as well. An similar approach can be used to obtain therapeutic nanobodies (ai-Nbs) against T3SS components of other pathogenic microorganisms. In a number of studies, anti-infective nanobodies were obtained that bind to pathogen surface structures, components of secretion or transport systems [46-50]. We hope that the ai-Nb-based method described in this paper could be applied for obtaining biomimetics of “sensitive” pathogenic epitopes of these nanobodies as well to be used for possible therapeutic directed immunomodulation of the host immune system.

  1. Consider discussing how the use of aiNbs as a component of a subunit vaccine compares to other vaccine approaches, especially in terms of efficacy, safety, and ease of administration.

 - We have added the following text to the Discussion section.

As a component of a subunit vaccine, aiNbs have all the useful futures of nanobodies in terms of the very efficient technologies developed for their generation, selection and formatting (modifications, multi-module constructs), of producing and administration, safety and a relatively simplified procedure for their humanization if required. The lifetime of a nanobody in the bloodstream in case of systemic administration can be adjusted in various ways 23-25].

  1. Discuss the potential impact of aiNbs on the field of monoclonal antibody (mAb) therapeutics. Consider how the immunogenicity of aiNbs may differ from traditional mAbs and how this might influence their clinical utility.

- At the end of the the Discussion section is written:

In modern terminology we assume the use of aiNbs for targeted immunomodulation or immunostimulation of immune cells using the engineered aiNbs or their derivatives. Modulating the immune system is a pivotal treatment strategy of modern medicine. Currently, one of the greatest challenges in mAb therapeutics (especially with systemic use of immunotherapeutic drugs) is their immunogenicity and the formation of anti-drug antibodies which decrease their clinical efficacy [51,52]. In the case of aiNbs, it is their immunogenicity that is important for the formation of host’s own therapeutic antibodies, and the systemic administration of the vaccine could potentially be replaced by local (subcutaneous).

The developed subunit vaccines have been found to be poorly immunogenic, and thus, multiple boosters and suitable adjuvantation are necessary to augment their protective potential [53].

Round 2

Reviewer 1 Report

Comments and Suggestions for Authors

The authors have responded to all the questions. I suggest it could be published in present form. 

Reviewer 2 Report

Comments and Suggestions for Authors

The manuscript reads fine now.